# Insights into metabolic heterogeneity of colorectal cancer gained from fluorescence lifetime imaging

Anastasia D Komarova[1,2†], Snezhana D Sinyushkina[1†], Ilia D Shchechkin[1,2†], Irina N Druzhkova[1], Sofia A Smirnova[1], Vitaliy M Terekhov[3], Artem M Mozherov[1], Nadezhda I Ignatova[1], Elena E Nikonova[4], Evgeny A Shirshin[4,5], Liubov E Shimolina[1], Sergey V Gamayunov[3], Vladislav I Shcheslavskiy[1,6], Marina V Shirmanova[1]*

[1]Institute of Experimental Oncology and Biomedical Technologies, Privolzhsky Research Medical University, Nizhny Novgorod, Russian Federation; [2]Institute of Biology and Biomedicine, Lobachevsky State University of Nizhny Novgorod, Nizhny Novgorod, Russian Federation; [3]Nizhny Novgorod Regional Oncologic Hospital, Nizhny Novgorod, Russian Federation; [4]Laboratory of Clinical Biophotonics, Sechenov First Moscow State Medical University, Moscow, Russian Federation; [5]Faculty of Physics, Lomonosov Moscow State University, Moscow, Russian Federation; [6]Becker&Hickl GmbH, Berlin, Germany

*For correspondence:
Shirmanovam@mail.ru

†These authors contributed equally to this work

**Abstract** Heterogeneity of tumor metabolism is an important, but still poorly understood aspect of tumor biology. Present work is focused on the visualization and quantification of cellular metabolic heterogeneity of colorectal cancer using fluorescence lifetime imaging (FLIM) of redox cofactor NAD(P)H. FLIM-microscopy of NAD(P)H was performed in vitro in four cancer cell lines (HT29, HCT116, CaCo2 and CT26), in vivo in the four types of colorectal tumors in mice and ex vivo in patients' tumor samples. The dispersion and bimodality of the decay parameters were evaluated to quantify the intercellular metabolic heterogeneity. Our results demonstrate that patients' colorectal tumors have significantly higher heterogeneity of energy metabolism compared with cultured cells and tumor xenografts, which was displayed as a wider and frequently bimodal distribution of a contribution of a free (glycolytic) fraction of NAD(P)H within a sample. Among patients' tumors, the dispersion was larger in the high-grade and early stage ones, without, however, any association with bimodality. These results indicate that cell-level metabolic heterogeneity assessed from NAD(P)H FLIM has a potential to become a clinical prognostic factor.

## eLife assessment

This study presents a **valuable** finding on the heterogeneity of tumour metabolism using fluorescence lifetime imaging, measured across 4 cell lines, 4 tumour types of in vivo mouse models, and 29 patient samples. The indication is that the level of heterogeneity of cellular metabolism increases with model complexity and demonstrates high heterogeneity at a clinical level. The evidence supporting the claims of the authors is **solid**, and at the revision stage, the authors have included additional samples from 8 patients in the data pool, which is helpful for the conclusions that the authors are trying to draw. The work will be of interest to medical biologists developing methods for quantifying metabolic heterogeneity.

## Introduction

Reprogramming of energy metabolism is an established hallmark of malignant tumors. Tumor cells adjust their metabolism to sustain uncontrolled proliferation and tumor progression, even in the conditions of low nutrient supply and hypoxia. There are two main metabolic pathways for energy production in the form of ATP – glycolysis and oxidative phosphorylation (OXPHOS). For a long time, enhanced glycolysis has been considered as a central feature of tumor metabolism (*DeBerardinis and Chandel, 2016*). Unlike most normal cells, tumor cells rely on glycolysis not only in hypoxia (anaerobic glycolysis), but also in normoxia (aerobic glycolysis, or the Warburg effect), which provides them with many metabolic intermediates and a high-speed production of ATP for fast growth. The glycolytic shift in the tumors traditionally correlates with negative prognosis (*Zhou et al., 2022*). At the same time, mitochondrial OXPHOS is as important in malignant cells as glycolysis. Although defective mitochondria and lower OXPHOS capacity are often observed in cancer cells, this is not an absolute phenomenon – many tumors preserve functional mitochondria and normal OXPHOS rate (*Gentric et al., 2017*). Along with glucose, tumors use glutamine and fatty acids as alternative substrates. Interestingly, tumor cells are capable of switching between different metabolic pathways depending on their own needs and local environment, thus demonstrating a high degree of metabolic plasticity.

Therefore, tumor metabolism is currently viewed as a dynamic, variable system with a diversity of cellular metabolic states. In this context, intratumor heterogeneity means that there are the cells with different metabolic profiles in a tumor concurrently, and intertumor heterogeneity means that tumors of the same type and stage are characterized by the different metabolic strategies (*Kim and DeBerardinis, 2019*).

Heterogeneity of tumor metabolism can be caused by various factors (*Shirmanova et al., 2023*). Some of them («intrinsic») are related to tumor cells themselves: these are histogenesis of the tumor, mutation profile, (epi)genetics factors, differentiation state and proliferation activity, to name a few (*Pavlova and Thompson, 2016*; *Sengupta and Pratx, 2016*; *Seth Nanda et al., 2020*; *Farhadi et al., 2022*). Other reasons («extrinsic») are associated with the nonuniform microenvironment, for example local hypoxia, nutrient distribution, heterogeneity of the vasculature network, interaction of tumor cells with the extracellular matrix and stromal cells, etc. (*Marusyk and Polyak, 2010*; *Masson and Ratcliffe, 2014*; *Yoshida, 2015*; *Stine et al., 2015*). Collectively, these factors lead to variability of tumor metabolism in space and time. Metabolic heterogeneity as a part of general heterogeneity of tumors imposes some difficulties in patients' diagnosis and treatment and is believed to have a prognostic value (*Liu et al., 2022a*; *Liu et al., 2022b*; *Pinho et al., 2020*).

Although metabolic heterogeneity is a well-recognized feature of tumors, its characterization at the cellular level remains scarce. In part, this is associated with the lack of the highly sensitive and direct techniques for its observation. Clinical imaging modalities, such as metabolic PET with radiolabeled tracers ($^{18}$F, $^{11}$C) and MRI/MRS, allow to capture inter- and intratumor heterogeneity among patients with low (~4.5 mm) spatial resolution (*Plathow and Weber, 2008*). Recently evolved methods of single-cell sequencing deliver comprehensive information about the metabolic landscape of a tumor with a resolution up to 55–100 μm based on the expression of the metabolic genes, but these approaches are rather complex, laborious and not widely available (*Evers et al., 2019*; *Huang et al., 2023*).

Interrogation of cellular metabolism is possible with the laser scanning fluorescence microscopy that enables the detection of autofluorescence of the redox cofactors, such as the reduced form of the nicotinamide adenine dinucleotide (phosphate) – NAD(P)H and oxidized flavins (*Georgakoudi and Quinn, 2023*). Recording of the fluorescence decay parameters using the option of the fluorescence lifetime imaging (FLIM) allows the evaluation of the states of the cofactor molecules attributed to different metabolic pathways. The unbound state of NAD(P)H has a short lifetime (~0.4 ns), while the protein-bound state has a long lifetime (~1.7–3.0 ns; *Lakowicz et al., 1992*). Changes in the relative fractions of the unbound and bound NAD(P)H in cancer cells calculated from biexponential fitting of decay curve typically indicate the shifts between glycolysis and OXPHOS (*Rück et al., 2014*). FLIM of NAD(P)H is currently considered as a promising, label-free technique for the assessment of metabolic heterogeneity of tumors at the (sub)cellular scale. Several studies, including ours, demonstrate the possibilities of NAD(P)H FLIM to visualize metabolic heterogeneity in cultured cells, multicellular structures in vitro, tumor xenografts in vivo and patients' tumors ex vivo (*Shah et al., 2015*; *Skala*

*et al., 2022*; *Lukina et al., 2019*). However, quantifying the degree of tumor heterogeneity is lacking in most of these works.

In this paper, we present the results of assessment of cell-level metabolic heterogeneity of colorectal cancer using FLIM-microscopy of NAD(P)H. The distributions of the fluorescence decay parameters have been analyzed in the four colorectal cancer cell lines (HT29, HCT116, CaCo2, CT26), in the mouse tumor models in vivo generated from these cell lines and in the post-operative samples of patients' colon tumors. Quantification of heterogeneity has been performed with the dispersion (D) and the bimodality index (BI) of the decay parameters. SHAP (SHapley Additive exPlanations) analysis has been conducted to find associations of the heterogeneity degree with clinical prognosis.

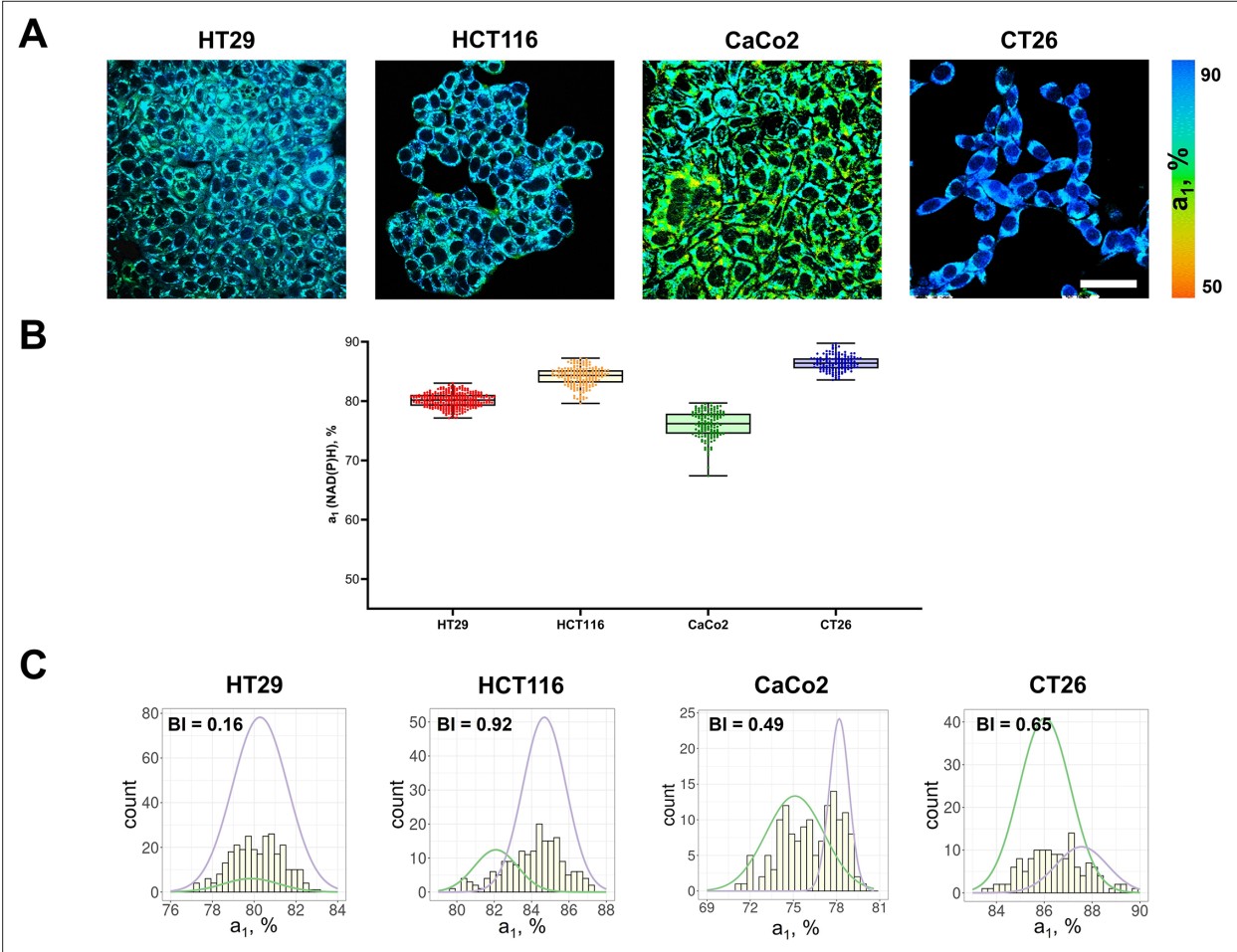

**Figure 1.** FLIM of NAD(P)H in monolayer cell cultures. (**A**) Representative FLIM images of colorectal cancer cell lines. Scale bar = 50 µm. For FLIM: ex. 750 nm, reg. 450–490 nm. (**B**) The relative contribution of free NAD(P)H ($a_1$, %) in cell cultures. Box shows the median and the quartiles Q1 and Q3, whiskers show minimum and maximum. Dots indicate individual cells (n=280 for HT29 cells, n=185 for HCT116 cells, n=146 for CaCo2 cells, n=138 for CT26 cells). p-values are shown in *Supplementary file 1*. (**C**) The distribution of the NAD(P)H-$a_1$ for the cell lines. The bimodality index (BI-$a_1$) is shown on each diagram.

The online version of this article includes the following source data and figure supplement(s) for figure 1:

**Source data 1.** Original FLIM data for *Figure 1A* (HT29 cells).

**Source data 2.** Original FLIM data for *Figure 1A* (HCT116 cells).

**Source data 3.** Original FLIM data for *Figure 1A* (CaCo2 cells).

**Source data 4.** Original FLIM data for *Figure 1A* (CT26 cells).

**Source data 5.** The dataset (NAD(P)H-$a_1$ values) used to plot the charts shown in *Figure 1B*.

**Figure supplement 1.** FLIM of NAD(P)H in monolayer cell cultures.

**Figure supplement 2.** Autofluorescence of cofactors FAD and NAD(P)H in cultured cells HT29, HCT116, CaCo2 and CT26.

**Table 1.** NAD(P)H fluorescence decay parameters of colorectal cancer cells in monolayer cultures in vitro and in mouse tumors in vivo.

| Cell line | $\tau_m$, ns | $\tau_1$, ns | $\tau_2$, ns | $a_1$, % | BI-$\tau_m$ |
|---|---|---|---|---|---|
| Cell lines in vitro | | | | | |
| HT29 | 0.80±0.05 | 0.39±0.03 | 2.47±0.17 | 80.11±1.34 | 1.05 |
| HCT116 | 0.71±0.05 | 0.40±0.02 | 2.39±0.20 | 84.13±1.58 | 0.90 |
| CaCo2 | 0.95±0.10 | 0.38±0.05 | 2.53±0.24 | 73.48±2.26 | 1.30 |
| CT26 | 0.57±0.06 | 0.35±0.03 | 2.03±0.22 | 86.48±1.28 | 1.28 |
| Tumors in vivo | | | | | |
| HT29 | 0.84 [0.81;0.90] | 0.46 [0.42;0.48] | 2.66 [2.54;2.76] | 81.54 [79.93;83.13] | 0.85±0.35 |
| HCT116 | 0.88 [0.85;0.92] | 0.47 [0.45;0.48] | 2.66 [2.56;2.77] | 80.61 [79.32;81.96] | 0.91±0.28 |
| CaCo2 | 1.02 [0.86;1.19] | 0.42 [0.38;0.50] | 2.91 [2.61;3.32] | 76.89 [74.06;78.52] | 1.51±0.71 |
| CT26 | 0.72 [0.67;0.78] | 0.39 [0.37;0.41] | 2.34 [2.21;2.50] | 82.12 [80.96;84.70] | 1.20±0.36 |

$\tau_m$ – mean lifetime,  $\tau_1$ – short lifetime component,  $\tau_2$ – long lifetime component, $a_1$ – relative contribution of the short lifetime component, BI-$\tau_m$ – bimodality index of the mean lifetime.

## Results

### Metabolic heterogeneity assessment in colorectal cancer cell lines

First, using FLIM of NAD(P)H, cellular metabolism was assessed in monolayer cultures of different colorectal cancer cell lines (*Figure 1A*). Typical values of NAD(P)H fluorescence lifetimes were registered for all cell lines – short $\tau_1$~0.39 ns and long $\tau_2$~2.03–2.59 ns (*Table 1*). Due to specifics of cellular metabolism, the relative contributions of the free ($a_1$) and bound ($a_2$) forms of NAD(P)H and, consequently, the mean lifetimes $\tau_m$ varied between the cell lines. The $a_1$ value decreased and $\tau_m$ increased in the following order: CT26 (~86%, 0.57 ns)>HCT116 (~84%, 0.71 ns)>HT29 (~80%, 0.80 ns)>CaCo2 (~73%, 0.95 ns) (*Figure 1B*, *Table 1*). A high NAD(P)H-$a_1$ (low $\tau_m$) is an indicator of glycolytic shift, which is typical for cells with intense proliferation in a monolayer culture, like CT26 and HCT116. CaCo2 cells with the lowest $a_1$ value had more OXPHOS-shifted metabolism, which can be associated with the expression of morphological and functional characteristics of mature enterocytes of normal small intestine by this cell line (*Lea, 2015*). All cell lines statistically differed from each other (for p-val, see *Supplementary file 1*), and intercellular variations of the $a_1$ parameter were minor (<3%).

To quantify the metabolic heterogeneity, the BI was calculated for distributions of both $a_1$ (*Figure 1C*, *Table 2*) and $\tau_m$ values (*Figure 1—figure supplement 1*, *Table 1*). For all cell lines the BI of the $a_1$ distribution did not exceed 1.1 (the threshold of bimodality, *Wang et al., 2009*), justifying the uniformity of cell metabolism in a culture, which is consistent with the general view on standard cell lines as homogenous populations (*Auman and McLeod, 2010*; *Idrisova et al., 2022*). The BI-$\tau_m$ was, however, rather high (>1.0) in all cell lines except HCT116 (0.90). In the further experiments on tumors the BI-$a_1$ was used as more relevant.

Additionally, the dispersion of NAD(P)H-a1 (D-a1) was evaluated for each cell line to describe the extent of distribution of the data around the median (Table 2). The D-a1 value varied from 1.67% in CT26 cell culture to 3.41% in CaCo2.

### Metabolic heterogeneity in mouse tumor models in vivo

Next, FLIM of NAD(P)H was performed in vivo on mouse tumor models, obtained from the colorectal cell lines shown above (*Figure 2A*). All the tumors were verified by histopathological analysis (*Figure 2B*).

Among the tumors, CT26 had the highest (~82%), and CaCo2 had, on average, the lowest (~76%, p-val=0.0028) contribution of free NAD(P)H $a_1$, similar to the cultured cells (*Figure 2C*, *Table 1*).

Most of the individual tumors showed larger (≥3%) inter-cellular variations of NAD(P)H-$a_1$ than corresponding cell lines, where the deviation from the median was ≤3%. The dispersion D-$a_1$ was in the range of 2.6–4.0% (*Table 2*). At that, intertumor differences across each type of tumor were insignificant.

**Table 2.** The bimodality index $BI\text{-}a_1$ and dispersion $D\text{-}a_1$ of NAD(P)H in cultured cells, mouse tumors and patients' tumor samples.

**Cell lines in vitro**

|           | HT29 | HCT116 | CaCo2 | CT26 |
|-----------|------|--------|-------|------|
| $BI\text{-}a_1$ | 0.16 | 0.92 | 0.49 | 0.63 |
| $D\text{-}a_1$ | 1.83 | 2.01 | 3.41 | 1.67 |

**Tumors in vivo**

|           | HT29 | HCT116 | CaCo2 | CT26 |
|-----------|------|--------|-------|------|
| $BI\text{-}a_1$ | 1.2±0.32 | 0.93±0.23 | 0.86±0.15 | 1.06±0.54 |
| $D\text{-}a_1$ | 3.20 | 2.60 | 4.00 | 2.81 |

**Patients' tumors ex vivo**

| Sample | 1 | 2 | 3 | 4 | 5 | 6 | 7 | 8 | 9 | 10 |
|--------|------|------|------|------|------|------|------|------|------|------|
| $BI\text{-}a_1$ | 1.25 | 0.81 | 1.01 | 1.16 | 1.01 | 1.00 | 0.86 | 0.70 | 1.28 | 1.25 |
| $D\text{-}a_1$ | 7.77 | 4.18 | 8.68 | 11.99 | 6.93 | 7.83 | 9.33 | 8.25 | 11.51 | 6.96 |

| Sample | 11 | 12 | 13 | 14 | 15 | 16 | 17 | 18 | 19 | 20 |
|--------|------|------|------|------|------|------|------|------|------|------|
| $BI\text{-}a_1$ | 0.59 | 1.45 | 1.34 | 1.05 | 0.89 | 0.24 | 1.29 | 1.32 | 0.80 | 0.94 |
| $D\text{-}a_1$ | 4.5 | 2.37 | 4.94 | 2.19 | 4.16 | 3.34 | 6.10 | 5.41 | 2.23 | 5.19 |

| Sample | 21 | 22 | 23 | 24 | 25 | 26 | 27 | 28 | 29 | |
|--------|------|------|------|------|------|------|------|------|------|--|
| $BI\text{-}a_1$ | 1.28 | 0.98 | 1.53 | 1.31 | 0.88 | 1.58 | 1.65 | 0.5 | 1.29 | |
| $D\text{-}a_1$ | 5.27 | 3.29 | 6.54 | 5.77 | 4.87 | 5.83 | 4.14 | 5.67 | 6.36 | |

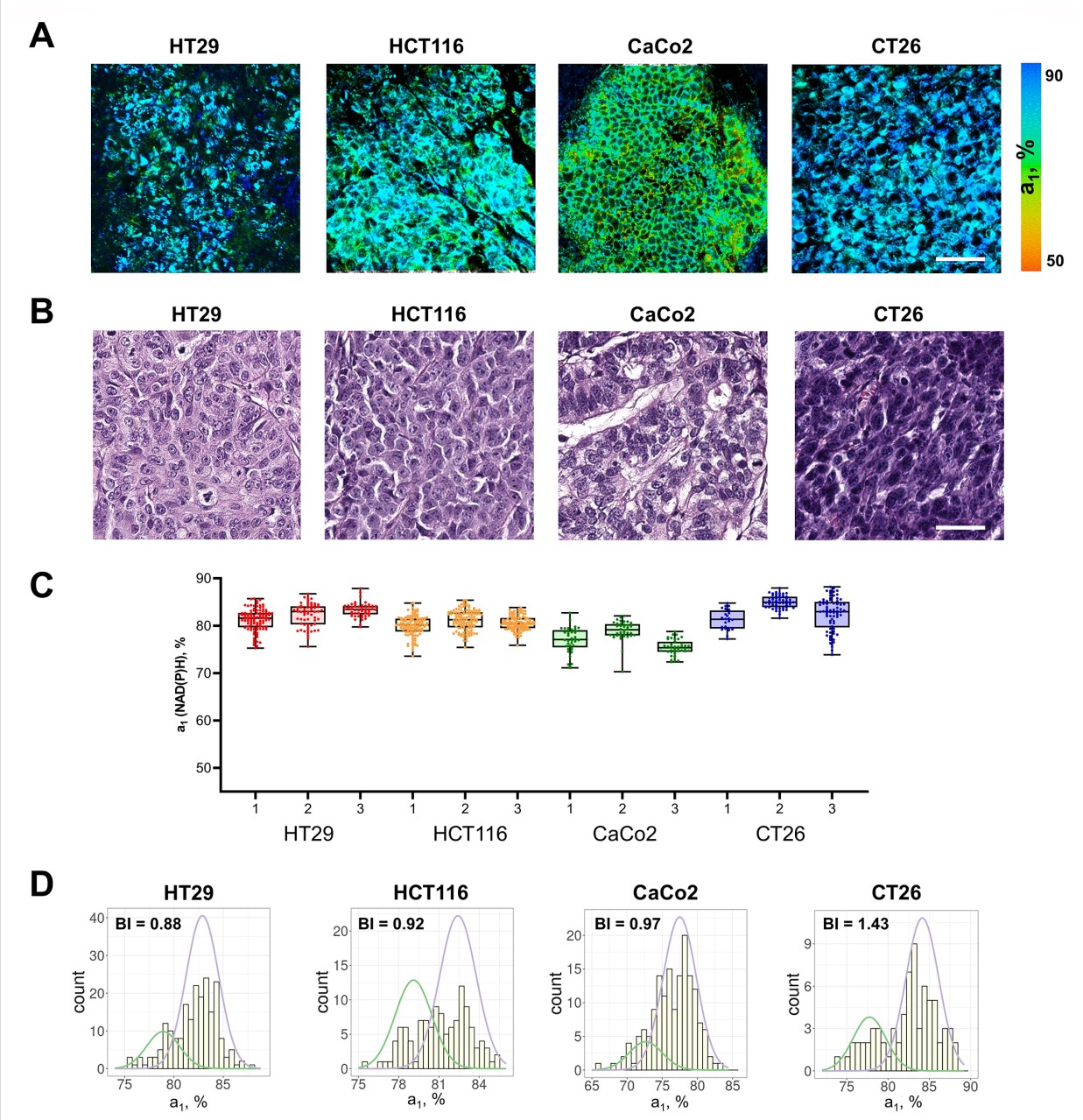

**Figure 2.** FLIM of NAD(P)H in mouse tumors in vivo. (**A**) FLIM images of NAD(P)H of tumor cells in mouse models in vivo. Scale bar = 50 μm. For FLIM: ex. 750 nm, reg. 450–490 nm. (**B**) Representative histological slices of tumors, hematoxylin/eosin (HE) staining, initial magnification 20×. Scale bar = 50 μm. (**C**) The relative contribution of free NAD(P)H ($a_1$, %) in three representative tumors (numbered 1–3) obtained from different cell lines. Box shows the median and the quartiles Q1 and Q3, whiskers show minimum and maximum. Dots indicate individual cells (n=280 for HT29, n=340 for HCT116, n=160 for CaCo2, n=350 for CT26). p-values are shown in ***Supplementary file 1***. (**D**) Representative distributions of the NAD(P)H-$a_1$ for each type of tumor. The bimodality index (BI-$a_1$) is shown on the diagrams.

The online version of this article includes the following source data and figure supplement(s) for figure 2:

**Source data 1.** Original FLIM data for ***Figure 2A*** (HT29 tumor).

**Source data 2.** Original FLIM data for ***Figure 2A*** (HCT116 tumor).

**Source data 3.** Original FLIM data for ***Figure 2A*** (CaCo2 tumor).

**Source data 4.** Original FLIM data for ***Figure 2A*** (CT26 tumor).

**Source data 5.** The dataset (NAD(P)H-$a_1$ values) used to plot the charts shown in ***Figure 2C***.

*Figure 2 continued on next page*

*Figure 2 continued*

**Figure supplement 1.** Immunohistochemical analysis of the expression of EpCAM (green, epithelial cells marker) and vimentin (red, mesenchymal cells marker) in mouse tumors.

**Figure supplement 2.** Autofluorescence of cofactors FAD and NAD(P)H in HT29 and HCT116 tumor xenografts in vivo.

The BI-$a_1$ values in mouse tumors were generally higher than in monolayer cultures (*Figure 2D*, *Table 2*). In 5 of 19 tumors the BI-$a_1$ was ≥1.1, indicating the bimodal distribution of the $a_1$ parameter, that is the presence of two subpopulations of cells with different metabolism. Ten tumors had the BI-$a_1$ in the range of 0.7–1.0, which suggests that, while the distribution of the estimated parameter was unimodal, it was either wide or asymmetric, thus, also indicating some degree of heterogeneity. In 4 of 19 tumors the BI-$a_1$ was small (0.43–0.69), suggesting uniformity of cells' metabolism. Given the genetic identity of cells in standard cell lines, we can assume that the nonuniform microenvironment in the tumors was a major source of their variable metabolism.

## Metabolic heterogeneity in colorectal cancer samples from patients

NAD(P)H FLIM images were collected from 29 postoperative samples of patients' colorectal adenocarcinomas, among which were the tumors of the I–IV stages, poorly and highly differentiated (*Table 3*, *Supplementary file 2*). The representative FLIM and histological images are presented in *Figure 3A and B* correspondingly. Patients' tumors showed fluorescence lifetimes of $\tau_1$~0.45 ns and $\tau_2$~1.80–3.20 ns (*Supplementary file 2*), which were comparable with the values in human tumor xenografts (*Table 1*). The parameter NAD(P)H-$a_1$ was in the range of ~62–80% and NAD(P)H-$\tau_m$ was in the range

**Table 3.** Information about patients and their colorectal tumors.

| Characteristics | Number | Percent |
|---|---|---|
| Gender | | |
| Male | 16 | 55.17 |
| Female | 13 | 44.83 |
| Age | | |
| Mean ±SD | 65.28±11.92 y.o. | - |
| Median | 67 y.o. | - |
| Tumor staging | | |
| I | 3 | 10.34 |
| IIA | 6 | 20.69 |
| IIB | 3 | 10.34 |
| IIIB | 11 | 37.93 |
| IIIC | 1 | 3.45 |
| IV | 5 | 17.25 |
| Tumor site | | |
| Cecum colon | 2 | 6.90 |
| Transverse colon | 10 | 34.48 |
| Hepatic flexure | 3 | 10.34 |
| Sigmoid colon | 8 | 27.59 |
| Rectum | 6 | 20.69 |
| Grade | | |
| Low (G1) | 4 | 13.79 |
| Moderate (G2) | 19 | 65.52 |
| High (G3) | 6 | 20.69 |

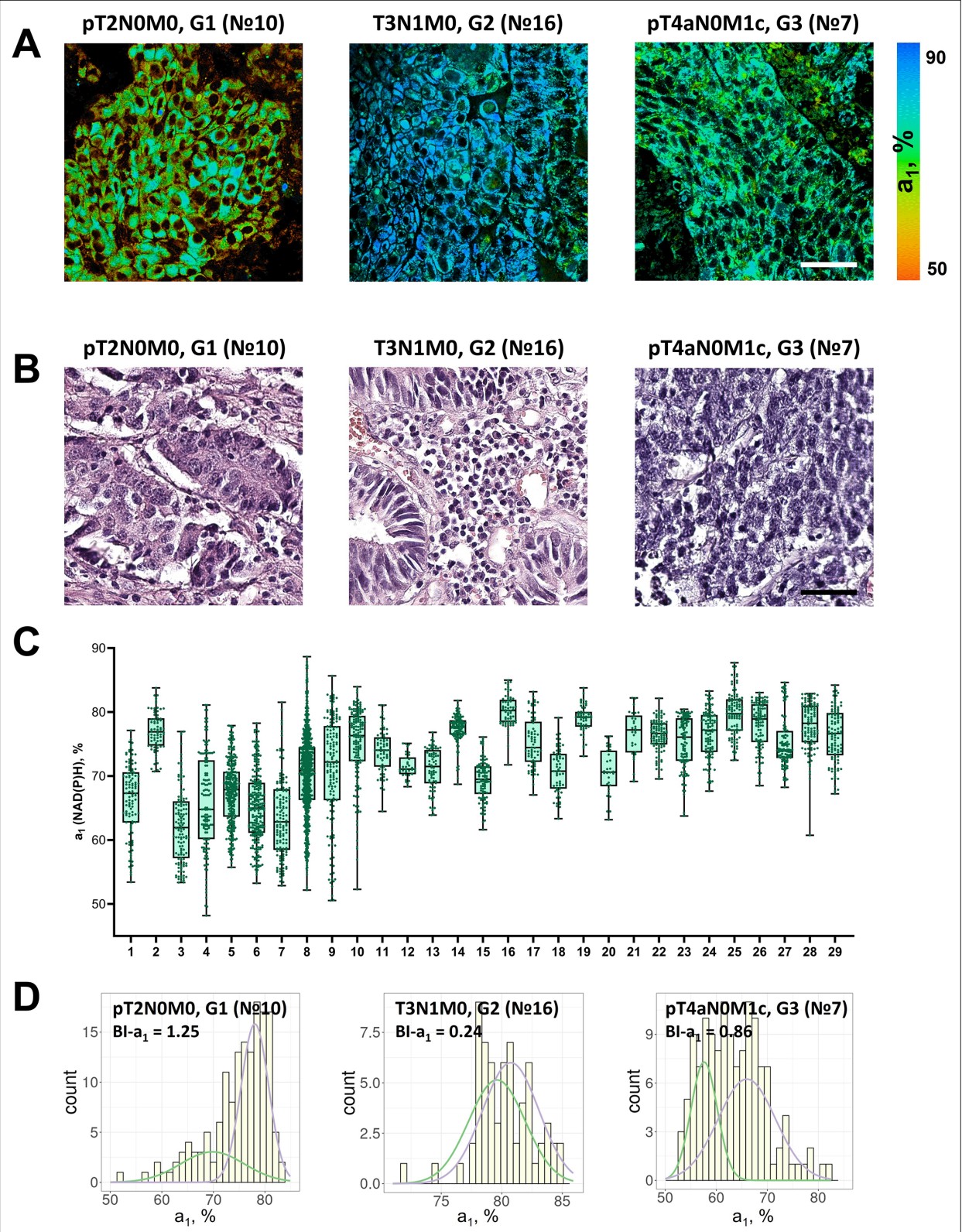

**Figure 3.** FLIM of NAD(P)H in patients' tumor samples ex vivo. (**A**) Representative FLIM images of patient tumors. Scale bar = 50 µm. For FLIM: ex. 750 nm, reg. 450–490 nm. (**B**) Histopathology of tumors, hematoxylin/eosin (HE) staining, initial magnification 20×. Scale bar = 50 µm. (**C**) The relative contribution of free NAD(P)H ($a_1$, %) in patients' tumors (numbered 1–29). Box shows the median and the quartiles Q1 and Q3, whiskers show minimum

*Figure 3 continued on next page*

*Figure 3 continued*

and maximum. Dots are the measurements from the individual cells. (**D**) Representative distributions of the NAD(P)H-$a_1$ for patients' tumors. The bimodality index (BI-$a_1$) is shown on the diagrams.

The online version of this article includes the following source data and figure supplement(s) for figure 3:

**Source data 1.** The dataset (NAD(P)H-$a_1$ values) used to plot the charts shown in *Figure 3C*.

**Figure supplement 1.** Immunohistochemical analysis of the expression of EpCAM (green, epithelial cells marker) and vimentin (red, mesenchymal cells marker) in patients' tumors.

**Figure supplement 2.** Immunohistochemical analysis of the expression of GLUT3 and LDHA in patients' tumors.

**Figure supplement 3.** Autofluorescence of cofactors FAD and NAD(P)H in patient tumor ex vivo (№ 11).

of 0.80–1.20 ns, indicating that patients' tumors generally had larger variability of metabolic statuses than cancer cells in vitro and in xenografts in vivo.

A high degree of inter- and intratumor variability of cellular metabolism was detected in patients' tumors (*Figure 3*). Less than half of the tumors (13 of 29) showed deviation of NAD(P)H-$a_1$ from the median <10% across the cells, and for the rest (16 of 29) the variations were 10–25%. The dispersion D-$a_1$ varied significantly among the samples, from 2.19% to 11.99%.

According to the heterogeneity assessment using the bimodality index, 14 of 29 tumors were metabolically highly heterogeneous (BI-$a_1$ ≥1.1) (*Figure 3D*, *Table 2*). In 14 tumors the BI-$a_1$ value was in the range of 0.50–1.0, which indicated the presence of metabolically different cells but not clearly separated into two clusters. Only one tumor sample (p16) was metabolically homogeneous (BI-$a_1$=0.24).

Notably, the bimodality index showed no correlation with dispersion. That is, among the samples there were those with bimodal distribution, but small dispersion of NAD(P)H-$a_1$ in a cell population (e.g. samples № 12, 13, 21), and vice versa (e.g. samples № 3, 7, 8).

Therefore, using NAD(P)H FLIM we have observed and quantified metabolic heterogeneity of patients' colorectal tumors. Unlike tumor xenografts obtained from the cell lines that are thought to be genetically stable and identical, patients' tumors are genetically diverse, which could also contribute to their metabolic heterogeneity, in addition to microenvironmental factors.

## Interrelation between metabolic heterogeneity and clinicopathological characteristics of tumors

The relationships between the metrics of cellular heterogeneity – the bimodality index BI and dispersion D of NAD(P)H fluorescence decay parameters – with the clinical parameters of the tumor, such as the stage according to the TNM system, and the grade (G), were analyzed (*Figure 4*). Due to the small sample size, all tumors were divided into two groups by each parameter: T1 +T2 and T3 +T4; N0M0 and all the others with metastases; G1 +G2 and G3.

SHAP analysis was used to estimate the importance of each variable (D-$\tau_2$, D-$a_1$, D-$\tau_m$, BI-$\tau_2$, BI-$a_1$, BI-$\tau_m$) coming from the biexponential decay curves for individual predictions. Fluctuations of $\tau_1$ (fluorescence lifetime of free NAD(P)H) were not included in the analysis because they do not have a rational biological interpretation.

The results showed that, among the variables, dispersion of $a_1$ had a major relative weight in the prediction of all the clinical characteristics studied.

The tumors of the advanced stages T3 and T4 were characterized by reduced dispersion of $a_1$ compared with early stages T1 and T2, indicating their lower heterogeneity (*Figure 4*). The stage of the tumor T was significantly associated with the value of D-$a_1$ (p-val=0.02).

If metastases were present (N and M were different from 0 in TNM), then those primary tumors showed a tendency to have lower D-$a_1$ compared with tumors for which metastases were absent (p-val=0.056).

The high-grade tumors displayed a higher value of dispersion D-$a_1$ than the low-grade ones (*Figure 4*). Mann-Whitney U test showed a statistically significant difference between the groups of different grades (p-val=0.04).

Other variables had no significant associations with clinicopathological parameters of the tumors and did not separate the groups of tumors reliably (see, for example, box-plots for BI-$\tau_m$ and BI-$a_1$ values, *Figure 4—figure supplement 1*).

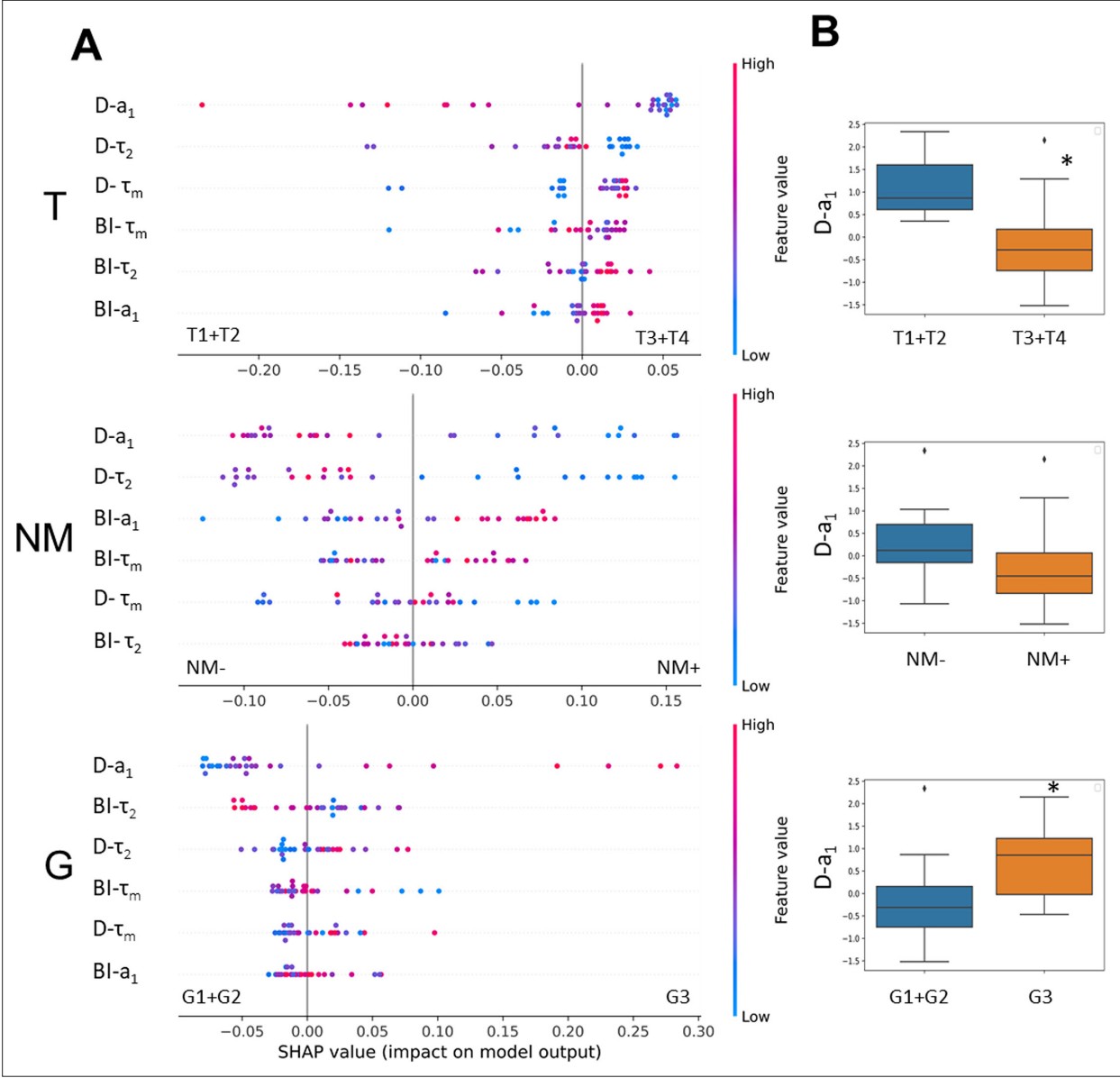

**Figure 4.** The relationships between metabolic heterogeneity and clinicopathological characteristics of patients' tumors. (**A**) Plots of SHAP analysis for the built decision tree models to determine the importance of dispersion (**D**) and bimodality index (BI) of the fluorescence decay parameters of NAD(P) H. The higher the value of the variable, the more red the dot is. (**B**) Box-plots of $D\text{-}a_1$ with highest significance, * p-val <0.05.

The online version of this article includes the following figure supplement(s) for figure 4:

**Figure supplement 1.** The relationships between parameters $BI\text{-}\tau_m$ (**A**) and $BI\text{-}a_1$ (**B**) and clinicopathological characteristics of patients' tumors.

Thus, the higher dispersion of free NAD(P)H fraction in a sample was characteristic of colorectal tumors of early stages (T1, T2) and high grade (G3) and thus showed a potential as a prognostic marker.

## Discussion

The altered energy metabolism is known to support tumor progression because tumor cells are critically in need of ATP for uncontrolled proliferation and growth. It has been established that tumor cells explore different metabolic programs and utilize multiple fuels even within one tumor, thus leading to metabolic heterogeneity. Until recently, direct observation of cell-level heterogeneity of tumor metabolism was a challenging task, but became possible with the evolution of advanced optical microscopic

techniques, such as two-photon fluorescence lifetime microscopy, FLIM. In the present research, using FLIM of redox cofactor NAD(P)H, which possesses endogenous fluorescence, we compared metabolic features of colorectal cancer cells across in vitro and in vivo models and patients' samples. For the first time, it was shown that heterogeneity of cellular metabolism increases with model complexity and is the highest at the level of patients' tumors. The heterogeneity was quantified on the basis of FLIM data and correlated with clinical characteristics of the tumors, which has not been done before.

A lot of studies, including the works of our group, demonstrate the possibilities of FLIM for assessment of the intra- and intertumor heterogeneity of metabolism in different models. Using NAD(P)H FLIM, metabolic heterogeneity was revealed in colorectal cancer cell cultures obtained from the patients' tumors, whereas the cells of the standard cell lines were uniform in their metabolism (*Shirshin et al., 2022*). J. Chacko and K. Eliceiri observed intercellular heterogeneity of metabolism in the MCF10A human breast epithelial cell line (*Chacko and Eliceiri, 2019*). Several studies show metabolic heterogeneity of 3D multicellular structures. For example, tumor spheroids obtained from the cervical cancer cell line HeLa (*Lukina et al., 2018*) or murine colorectal carcinoma cell line CT26 (*Shirmanova et al., 2021*) displayed the differences in metabolism between the outer and the inner cell layers with the outer layers being more glycolytic (higher NAD(P)H-$a_1$). The spheroids generated from the patients' derived glioblastoma cultures did not show metabolic zonality, but generally had greater intercellular variations of NAD(P)H lifetime parameters than the spheroids from standard line U373 MG (*Yuzhakova et al., 2023*). A spectrum of works by M. Skala' group demonstrates an opportunity of the optical metabolic imaging for heterogeneity assessment in patient tumor-derived organoids on the example of a breast, pancreatic (*Sharick et al., 2020*; *Walsh et al., 2014*; *Walsh et al., 2017*), head and neck (*Shah et al., 2017*), and colorectal cancers (*Skala et al., 2022*).

In the research in vivo, the heterogeneous structure of tumor and its microenvironment was shown on PyMT mammary tumors (*Burkel et al., 2022*). Analysis of the optical metabolic parameters revealed heterogeneity of the B78 mouse melanoma model, caused by different microenvironments (*Heaton et al., 2023*). The authors suggested that tumor heterogeneity could be induced by such factors as pH, nutrient and oxygen availability, or activation of immune cells. In the in vivo study on HeLa tumor xenografts, metabolically different cells were detected in the cellular and collagen-rich areas (*Shirmanova et al., 2018*). Recently, we demonstrated a high variability of cellular metabolic statuses in CT26 tumors of large size compared with small tumors, which correlated with heterogeneous oxygen distribution (*Parshina et al., 2022*). A significant intertumor heterogeneity of metabolism was observed in patient-derived glioblastoma xenografts, which differed them from standard U87 glioma (*Yuzhakova et al., 2022*).

In our study, we compared metabolic FLIM parameters of colorectal cell lines, mouse tumor models and patients' colorectal tumors. As expected, intercellular heterogeneity of metabolism was the highest in patients' tumor samples as followed from a wide and often bimodal distribution of NAD(P)H-$a_1$. This result is consistent with the study by J. Auman and H. McLeod using genome-wide gene expression data; the authors concluded that colorectal cancer cell lines lack the molecular heterogeneity of clinical colorectal tumors (*Auman and McLeod, 2010*).

Cell-level metabolic heterogeneity may be present in the cell population initially, as discussed above, or develop during the treatment. The heterogeneous metabolism after treatment was detected in tumor organoids (*Walsh et al., 2016*; *Walsh et al., 2014*; *Gillette et al., 2021*) and in animal models (*Heaster et al., 2019*). In any case, the presence of metabolically distinct cells led to the varying drug responses.

An assessment of metabolic heterogeneity in patients' tumors seems valuable from a clinical point of view. The great efforts to translate FLIM in the clinical setting both in vivo and ex vivo have been done by several groups (*Jo et al., 2018*; *Alfonso-Garcia et al., 2023*; *Weyers et al., 2022*; *Seidenari et al., 2012*; *Mycek et al., 1998*; *Herrando et al., 2024*; *Lagarto et al., 2020*). Clinical translation is spearheaded through macroscopic implementation and point-spectroscopy approaches that are capable of large sampling areas and enable access to otherwise constrained spaces but lack cellular resolution, making the interpretation of the results a complicated task. Previously, using NAD(P)H FLIM-microscopy of postoperative samples, we observed a high degree of inter- and intra-tumor heterogeneity of T3 stage colorectal tumors in comparison with normal colon samples (*Lukina et al., 2019*). Unfortunately, most of the tumor types are inaccessible for microscopic investigation in patients using clinically approved techniques. The exception is the skin tumors that can be inspected

with clinical systems, like MPTflex or MPTcompact (JenLab, Germany). At the same time, endoscopic FLIM-microscopy has been developing actively since the last ten years, which opens the prospects for in vivo examination of a wide spectrum of patients' tumor types (*Sun et al., 2013*; *Farhadi et al., 2022*). In principle, an assessment of tumor metabolism in biopsy samples is also possible, which widens the potential clinical applications of FLIM.

Although metabolic heterogeneity at the cellular level has been reported for different models and patient-derived cells, the comparisons between different samples and studies are hard to make as most of them lack any quantification of the heterogeneity.

Different approaches have been proposed to quantify the metabolic differences identified by FLIM. For example, in the papers (*Shah et al., 2015*; *Sharick et al., 2019*) a heterogeneity index and its modified form the weighted heterogeneity index (wH-index) were used for quantitative analysis of cellular heterogeneity. The wH-index is based on the Gaussian distribution models and is a modified form of the Shannon diversity index. Using this index the authors described the variations of a combination of parameters, such as NAD(P)H fluorescence lifetime, FAD fluorescence lifetime and optical redox ratio, defined as the OMI-index. The methods developed in *Heaster et al., 2019* established the combination of optical metabolic imaging variables and spatial statistical analysis (spatial proximity analysis, spatial clustering, multivariate spatial analysis, spatial principal components analysis) to quantify the spatial heterogeneity of tumor cell metabolism. Recently, we have proposed a new quantitative criterion – the bimodality index (BI) – to accurately discriminate between metabolically diverse cellular subpopulations on the basis of NAD(P)H FLIM data (*Shirshin et al., 2022*). The BI provides dimensionless estimation on the inherent heterogeneity of a sample by checking the hypothesis about approximation of a fluorescence decay parameter distribution by two Gaussians. Using the BI, the metabolic heterogeneity has been identified in standard and primary cancer cells cultures after chemotherapy with 5-fluorouracil.

Here, we used the dispersion of NAD(P)H $D-a_1$ and bimodality index $BI-a_1$ for quantifying metabolic heterogeneity of patient's tumors and compared it with in vitro and in vivo models. Of these two metrics of heterogeneity, the dispersion appeared more valuable in terms of tumor prognosis. Our results on patients' colorectal tumors revealed some associations between the dispersion of $a_1$ within a sample and tumor stage – the early-stage tumors (T1, T2) were metabolically more heterogeneous than the late-stage ones (T3, T4). A degree of heterogeneity was also associated with differentiation state, a stage-independent prognostic factor in colorectal cancer where the lower grade correlates with better the prognosis. The high-grade (G3) tumors had significantly higher dispersion of $a_1$, compared with low-grade ones (G1, G2). These results have a rational explanation from the point of view of biological significance of heterogeneity. In stressful and unfavorable conditions, to which the tumor cells are exposed, the spread of the parameter distribution in the population rather than the presence of several distinct clusters (modes) matters for adaptation and survival. The high diversity of cellular metabolic phenotypes provided the survival advantage, and so was observed in more aggressive (undifferentiated or poorly differentiated) and the least advanced tumors.

One of the possible reasons for metabolic heterogeneity could be the presence of stromal cells or diversity of epithelial and mesenchymal phenotypes of cancer cells within a tumor. Immunohistochemical staining of tumors for EpCam (epithelial marker) and vimentin (mesenchymal marker) showed that the fraction of epithelial, EpCam-positive, cells was more than 90% in tumor xenografts (*Figure 2— figure supplement 1*) and on average 76 ± 10% in patients' tumors (*Figure 3—figure supplement 1*). However, the ratio of EpCam- to vimentin-positive cells in patients' samples neither correlated with $D-a_1$ nor with $BI-a_1$, which means that the presence of cells with mesenchymal phenotype did not contribute to metabolic heterogeneity of tumors identified by NAD(P)H FLIM.

Metabolic heterogeneity of colorectal cancer is discussed in the literature. The focus of many of these studies is on the molecular classification of tumors by the analysis of their metabolic features. For example, *Zhang et al., 2020* showed that colorectal tumors could be classified into three distinct metabolism-relevant subtypes and developed a metabolism-related signature consisting of 27 metabolic genes, which were expressed differentially among the three subtypes and correlated with patients' overall survival . *Varshavi et al., 2020* performed metabolic characterization of colorectal cancer cells depending on KRAS mutation status using $^1H$ NMR spectra of the metabolites. They revealed that some mutations led to an increase of glucose consumption and lactate release, while others, on the contrary, decreased it. Numerous studies have investigated the prognostic potential of

key metabolic (mainly glycolytic) enzymes assessed from immunohistochemistry (IHC). For example, glucose transporter 3 (GLUT3) was highly expressed in colorectal cancer tissues of 63% of patients as relative to benign tissues and correlated with poor clinical outcomes (*Dai et al., 2020*). In the study by *Offermans et al., 2022* a sum score based on the expression levels of six proteins (PTEN, p53, GLUT1, PKM2, LDHA, MCT4) in colorectal cancer showed worse survival of patients with a higher probability of the Warburg effect . *Mizuno et al., 2020* found that expression of lactate dehydrogenase A (LDHA) at the invasive margin of the tumor was weaker than in the center. Note that standard molecular genetics and immunolabeling techniques identify the metabolic differences between tumors or between large areas within a tumor, while a single cell level metabolic information is lacking. We have verified the inability of IHC to detect intercellular differences in metabolic states based on the expression of LDHA and GLUT3. One can see from representative IHC images that the expression level of glycolytic enzymes provides some information about intertumor metabolic differences, but is insensitive to intercellular variability of metabolism registered by NAD(P)H FLIM (*Figure 3—figure supplement 2*).

Liu et al. found a correlation between intratumor metabolic heterogeneity parameters of $^{18}$F-FDG PET/CT and KRAS mutation status in colorectal cancer – KRAS mutant tumors had more $^{18}$F-FDG uptake and heterogeneity than wild-type KRAS (*Liu et al., 2022a*). In a different study they showed that intratumor metabolic heterogeneity assessed from $^{18}$F-FDG PET/CT is an important prognostic factor for progression-free survival and overall survival in patients with colorectal cancer (*Liu et al., 2022b*). The value of intratumoral metabolic heterogeneity in $^{18}$F-FDG PET/CT for prediction of recurrence in patients with locally advanced colorectal cancer was also demonstrated by *Han et al., 2018*. In the study by Zhang et al. intratumoral metabolic heterogeneity derived from $^{18}$F-FDG PET/CT was higher in colorectal tumors with a high microsatellite instability (MSI) – an important prognostic biomarker, and predicted MSI in stage I–III colorectal cancer patients preoperatively (*Zhang et al., 2023*). *Lin et al., 2021* identified two subtypes of colorectal cancer using a metabolic risk score based on genes that were mostly involved in lipid metabolism pathways; this criterion was applied for survival prediction – patients with a higher metabolic risk score had worse prognosis. Therefore, these clinical studies consider intratumor metabolic heterogeneity as a useful prognostic factor.

In the context of metabolic heterogeneity assay using NAD(P)H FLIM, some limitations should be mentioned. If the investigation of patients' tumor metabolism is performed on ex vivo tissue samples, one should be careful to work with freshly excised tissue or store the sample in the appropriate conditions. As we found previously, FLIM parameters of autofluorescence change very quickly (within 15 min) on air, but can be preserved in 10% BSA on ice during 3 h (*Lukina et al., 2019*). A limitation of our study is that microscopic FLIM images of the tissues were processed manually to obtain information about each individual cell, which made the analysis time-consuming. The methods for segmentation of tissue images have been continuously developed in digital histopathology (*Ahmed et al., 2022*; *van der Laak et al., 2021*). As for FLIM, there are approaches to automatic segmentation of FLIM images of cell cultures and some normal tissues on the basis of machine learning (*Smith et al., 2019*). So, in spite of the complex and heterogeneous tissue architecture, there are expectations that the task of identification of the individual cells in FLIM images of tumor tissues will be solved in the nearest future. Finally, a small size of the microscopic field of view (typically less 300 μm) limits the inspected area within the tumor sample. Taking into account a high spatial heterogeneity of clinical tumors, there is no confidence whether the inspected area is sufficiently representative.

Fluorescence of flavins can also serve as a biomarker of cellular metabolism independently of or in conjunction with NAD(P)H, for example in the optical redox ratio, OMI or FLIRR indexes (*Walsh et al., 2014*; *Wallrabe et al., 2018*; *Kalinina et al., 2021*). With the aim to assess metabolic heterogeneity in colorectal cancer, we have made an attempt to record the signal from flavins in addition to NAD(P)H. *Figure 1—figure supplement 2*, *Figure 2—figure supplement 2* and *Figure 3—figure supplement 3* show the examples of fluorescence images of flavins and NAD(P)H in colorectal cancer cell cultures and tissues. However, the fluorescence intensity of flavins was very low (~27 times lower than NAD(P)H) and insufficient to collect the required number of photons for correct fitting. Yet, this observation does not exclude that flavins can be useful in the heterogeneity assays of cancer of different origin.

Owing to the high (sub)cellular resolution (~200–500 nm), FLIM-microscopy provides unique information about metabolic heterogeneity, unavailable with any other methods, like $^{18}$F-FDG PET/CT, spatial transcriptomics or immunohistochemistry. Therefore, FLIM-microscopy of endogenous

cofactors is not only a powerful research technique, which is capable of improving our understanding of cellular metabolic diversity, but also the tool with a great potential for clinical translation to predict disease outcome.

## Conclusions

It is now evident that metabolic processes in cancer are highly variable, which make the tumor metabolism extremely heterogeneous. Metabolic heterogeneity of tumors complicates treatment efforts and is thought to be a negative prognostic factor. Due to the lack of methods for direct observation and quantification of metabolic heterogeneity at cellular level, it has been poorly characterized so far. Our assessments of cell-level heterogeneity from FLIM-microscopy of NAD(P)H clearly show that heterogeneous metabolic landscape of a patient' tumor is hardly reproducible in in vitro and in vivo models, which underlie the importance of such investigations on clinical material. Although the present research included a limited number of patients (n=29), the obtained results showing the associations between metabolic heterogeneity and clinical features of tumors allow us to consider it as a potential prognostic marker. We plan to continue this study to collect more tumor samples and estimate the correlations between their metabolic heterogeneity and follow-up clinical outcomes.

# Materials and methods

## Cell cultures

The human colorectal carcinoma cell lines H T 29, HCT116, CaCo2 and murine colon carcinoma cell line CT26 were used in the study. Cell lines were obtained from the Cell Culture Collection of Ivanovsky Institute of Virology, Gamaleya National Research Center of Epidemiology and Microbiology (Moscow, Russia), and their identity has been authenticated using STR profiling. Cells were grown in Dulbecco's modified Eagle's medium (DMEM; Gibco, Carlsbad, CA, USA) supplemented with 10% fetal bovine serum (FBS; Gibco, Carlsbad, CA, USA), 2 mM glutamine (Gibco, Carlsbad, CA, USA), 10 mg/mL penicillin (Gibco, Carlsbad, CA, USA), 10 mg/mL streptomycin (Gibco, Carlsbad, CA, USA). The cells were incubated at 37 °C, 5% $CO_2$, and 80% relative humidity and passaged three times a week. The passaging of cells was carried out at a confluence of 70–80% with trypsin-EDTA (Thermo Fisher Scientific, Waltham, MA, USA). Prior to use, the cell lines were tested for mycoplasma using PCR analysis and found to be uncontaminated.

For fluorescence microscopic studies the cells were seeded in 35 mm glass-bottomed FluoroDishes (Ibidi GmbH, Gräfelfing, Germany) in the amount of $5 \times 10^5$ cells in 2 mL of DMEM and incubated for 24 h (37 °C, 5% $CO_2$). Before FLIM, DMEM was replaced with the FluoroBrite DMEM (Thermo Fisher Scientific, Waltham, MA, USA) for fluorescence imaging.

Cell culture experiments included two independent replicates for each cell line, the data from which were then combined.

## Tumor models

In vivo experiments were performed on female nude and Balb/C mice weighing ~20–22 g purchased from a certified SPF vivarium of the Lobachevsky State University (Nizhny Novgorod, Russia). The animal study was conducted according to the guidelines of the Declaration of Helsinki and approved by the Local Ethical Committee of the Privolzhsky Research Medical University (approval № 09 from 30.06.2023). All surgery was performed under anesthesia, and every effort was made to minimize suffering. To generate tumors, the suspensions of cancer cells were injected subcutaneously into the tight in the following amount: $1.5 \times 10^6$ of H T 29 cells in 100 µL PBS, $1.0 \times 10^6$ of HCT116 cells in 100 µL PBS, $5.5 \times 10^6$ of CaCo2 cells in 100 µL PBS. CT26 cells ($2.0 \times 10^5$ cells in 20 µL PBS) were inoculated intracutaneously in the ear. The tumor volume was measured using a caliper, and calculated using the formula v=a × b×1/2b, where a is the length and b is the width of the tumor. In vivo studies were done on 21st day of growth for HT29 tumors (244.8±36.6 mm³), on the day 23 for HCT116 (533.9±42.0 mm³), on the day 68 for CaCo2 (397.8±65.2 mm³), and on the day 14 for CT26 (60.9±5.6 mm³). The groups of mice with HT29 and HCT116 tumors included three animals, CaCo2 - four animals, and CT26 - nine animals.

Mice were anesthetized by an intramuscular injection of Zoletil (40 mg/kg, Virbac SA, France) and 2% Rometar (10 mg/kg, Spofa, Czech Republic) for intravital microscopy. The skin over the tumor was opened and the tumor was covered with a coverslip.

After in vivo study, the mice were euthanized by cervical dislocation and tumors were surgically removed for histological analyses.

## Patient samples

Twenty-nine surgical samples of patients' colorectal tumors were obtained in the Nizhny Novgorod Regional Oncological Center (Nizhny Novgorod, Russia) during the tumor resection. The study with the use of patients' material was approved the Local Ethical Committee of the Privolzhsky Research Medical University (approval № 09 from 30.06.2023). All the patients gave informed written consent and consent to publish prior to the enrollment in the study.

All the patients had a histological verification of colorectal adenocarcinoma, the stage definition according to the TNM system, and the definition of the grade. There were tumors of the I–IV stage of low (G1), moderate (G2), and high (G3) grade. The tumors were localized in different sites of the large intestine (caecum, colon, rectum). Twenty-seven of 29 patients did not receive any anti-cancer therapy before the surgery, two patients (7 and 8) have been pretreated with radiotherapy or chemo-therapy. Patients summary is presented in *Table 3*, data about their clinicopathological characteristics is presented in *Supplementary file 2*.

Immediately after surgical excision, tumor samples, 0.5–1 $cm^3$ in size, were wrapped in gauze soaked in a solution of 10% BSA (bovine serum albumin), placed in sterile Petri dish on ice, transferred to the laboratory within 30 min and examined on the laser scanning microscope immediately. This storage protocol allows to preserve autofluorescence lifetime parameters unchanged for at least 3 h (*Lukina et al., 2019*).

## FLIM-microscopy

FLIM of NAD(P)H was performed using the laser scanning microscope LSM 880 (Carl Zeiss, Germany) equipped with a TCSPC-based FLIM module (Becker & Hickl GmbH, Germany). The Ti:Sa femtosecond laser MaiTai HP (Spectra-Physics Inc, USA, repetition rate 80 MHz, pulse duration 120 fs) was used for two-photon excitation of NAD(P)H at a wavelength of 750 nm. Fluorescence signal was registered in the range 450–490 nm. The laser power applied to the samples was ~6 mW. FLIM images were obtained using the water-immersion objective C-Apochromat W Korr 40×/1.3 (Carl Zeiss, Germany). Image collection time was 60 s. In total, 5–10 images were obtained from each sample.

FLIM images were processed in the SPCImage 8.5 software (Becker & Hickl GmbH, Germany). NAD(P)H fluorescence was analyzed in the cytoplasm of individual cells, in total 50–200 cells in each sample. Fluorescence decay curves were fitted by a bi-exponential model with a goodness of fit $\chi^2$ 0.8–1.2. The following fluorescence decay parameters were analyzed: short component corresponding to the free form of NAD(P)H ($\tau_1$), long component corresponding to the protein-bound NAD(P)H ($\tau_2$), their relative contributions ($a_1$ и $a_2$, correspondingly, $a_1+a_2$=100%), and the mean fluorescence lifetime ($\tau_m = \frac{a_1 \times \tau_1 + a_2 \times \tau_2}{a_1 + a_2}$).

## Histopathology and immunohistochemistry (IHC)

Formalin-fixed tumor samples were embedded in paraffin according to standard procedure and cut parallel to the optical plane. The sequential sections 7 µm thick were stained with hematoxylin and eosin, sections 4 µm thick were used for immunohistochemical staining.

Tissue sections were stained with primary antibodies to epithelial cell adhesion molecule EpCAM (Thermo Fisher Scientific, mouse, cat. #14-9326-82, dilution 1:700) and an intermediate filament protein vimentin (Abcam, ab137321, rabbit, dilution 1:700), according to the manufacturer's protocol. Secondary antibodies goat anti-mouse (Alexa Fluor 488) and goat anti-rabbit (Alexa Fluor 555) were used. Tissue sections were imaged using EVOS M7000 Imaging System (Thermo Fisher Scientific Inc, Waltham, MA USA) with LED cubes GFP (ex.470/22 nm, em.510/42 nm for Alexa 488) and RFP (ex.531/40 nm, em. 593/40 nm for Alexa 555) at x20 magnification.

Immunohistochemical staining for LDHA and GLUT3 was performed using immunostainer «Bond-Max» (Leica Biosistems, UK) with BOND 5.1 software, according to the standard protocols recommended by the manufacturer. Staining protocol included preliminary dewaxing of the sections and unmasking in a high pH buffer based on ethylenediaminetetraacetic acid for 20 min at 98–99°C. Next, slides were incubated with primary polyclonal antibodies to Glucose Transporter 3 GLUT3 (E-AB-31557, Elabscience, China) or to Lactate dehydrogenase A LDHA (E-AB-19947, Elabscience,

China) for 15 min. For the antibodies detection «Bond polymer refine detection system» (Leica Biosystems, UK) was used. The images were obtained using 3DHISTECH PANNORAMIC Midi (Carl Zeiss, Germany) at magnifications ×5, ×10, ×20, ×40, 63 x, resolution 0.087 µm/pixel. Due to the expression of GLUT3 and LDHA in all cell types and, especially, in cancer cells, the 100% of cells within the samples had positive staining. The staining intensity was visually evaluated as negative (-) low (+), moderate (++), or high (+++).

## Bimodality index and dispersion calculation

A continuous measure known as the «bimodality index» is utilized to gauge the degree of conformity of a set of univariate data to a two-component mixture model. The score is larger if the two components are balanced in size or if the separation between the two modes is larger. The BI ≥1.1 is considered as a cutoff to reliably define a bimodal distribution in the data (*Wang et al., 2009*; *Shirshin et al., 2022*).

BI was calculated according to the equation:

$$BI = \delta \times \sqrt{p\left(1 - p\right)}, \delta = \frac{|\mu_1 - \mu_2|}{\sqrt{\sigma_1 \sigma_2}}, p = \frac{n_1}{n_1 + n_2},$$

where µ – the mean of each Gaussian, σ – the standard deviation of each Gaussian and n – the number of measurements of each Gaussian.

Dispersion was calculated as interquartile range: 75% quantile – 25% quantile (Q3 – Q1).

The parameters $\tau_m$ and $a_1$ obtained by FLIM microscopy were used to calculate the bimodality indices and the dispersion for cultured cells, mouse tumors and patients' samples.

The bimodality index for $a_1$, $\tau_m$ and $\tau_2$ and dispersion of $a_1$, $\tau_m$ and $\tau_2$ were used to search for the relationships between them and the pathophysiological parameters of tumors: the TNM stage and the grade (G).

The fluorescence decay parameters were normalized using the z-normalization method to level out the numerical difference.

A random forest decision tree model was used to evaluate the importance of fluorescence decay parameters for clinicopathological characteristics. The model was used for feature selection analysis, by which the most significant ones are selected among the variables that define the sample. The random forest method delimited the variable space by attempting to reduce the Gini index as much as possible and partition the space into blocks containing only one type of sample. The model results were used for SHAP (SHapley Additive ExPlanations) analysis, which is designed to evaluate each variable for its ability to delimit space. SHAP analysis identified the most significant parameters, which were studied by classical statistical method (Mann-Whitney).

SHAP analysis is a technique used to explain the output of any machine learning model. SHAP analysis displays the relative influence of variables (decay parameters derivatives) on a model's output and can provide insight into the most significant factors and the impact of variables on the outcome. SHAP values are a way to explain the contribution of each object to the model output. They measure how much each variable contributes to the model's prediction and can help determine which variables are most important to the model and how they affect the outcome.

The T parameter from TNM was modified in a way that classes 1+2 became group 0, and 3+4 became group 1. The modification of the parameter G resulted in two classes: classes 1+2 became group 0 and class 3 became group 1. Parameters N and M were modified as NM, in which the class was distinguished: no metastases – 0, metastases – 1. Since the clinical parameters, in a modified form, are binary, a point-biserial correlation coefficient was chosen to assess the correlation.

## Statistical analysis

The obtained data were checked for the normal distribution using the Kolmogorov–Smirnov's criterion. The data with a normal distribution were presented as mean ± standard deviation (SD). The data with an abnormal distribution were presented as a median and 25% and 75% quartiles (Q1 and Q3). The ANOVA test for comparison data with a normal distribution and the Kruskal-Wallis's test for comparison data with an abnormal distribution were used, with p-val <0.05 being considered statistically significant. Statistical data processing was performed in IBM SPSS Statistics 26.0, R-studio, Python.

## Acknowledgements

The authors are grateful to Dr. Anton Plekhanov (PRMU) and Dr. Maria Lukina for helpful discussions and valuable suggestions and to Daria Myalik (PRMU) and Artem Grishin (PRMU) for their help with immunohistochemistry.

## Additional information

### Competing interests

Vladislav I Shcheslavskiy: The author was affiliated with Becker&Hickl GmbH at the time of research. The author has no other competing interests to declare. The other authors declare that no competing interests exist.

### Funding

| Funder | Grant reference number | Author |
| --- | --- | --- |
| Russian Science Foundation | 23-15-00294 | Anastasia D Komarova<br>Snezhana D Sinyushkina<br>Irina N Druzhkova<br>Elena E Nikonova<br>Evgeny A Shirshin<br>Vladislav I Shcheslavskiy |

The funders had no role in study design, data collection and interpretation, or the decision to submit the work for publication.

### Author contributions

Anastasia D Komarova, Snezhana D Sinyushkina, Validation, Investigation, Visualization, Writing – original draft; Ilia D Shchechkin, Data curation, Validation, Visualization, Methodology, Writing – original draft; Irina N Druzhkova, Formal analysis, Funding acquisition, Investigation; Sofia A Smirnova, Nadezhda I Ignatova, Liubov E Shimolina, Investigation; Vitaliy M Terekhov, Resources, Investigation; Artem M Mozherov, Data curation, Investigation, Methodology; Elena E Nikonova, Investigation, Methodology; Evgeny A Shirshin, Formal analysis, Methodology, Writing – original draft; Sergey V Gamayunov, Resources, Data curation; Vladislav I Shcheslavskiy, Data curation, Formal analysis, Validation, Methodology, Writing – original draft, Writing – review and editing; Marina V Shirmanova, Conceptualization, Data curation, Supervision, Writing – original draft, Writing – review and editing

### Author ORCIDs

Snezhana D Sinyushkina ⓘ https://orcid.org/0000-0001-9325-1746
Marina V Shirmanova ⓘ https://orcid.org/0000-0002-3207-7227

### Ethics

All the patients gave informed written consent and consent to publish prior to the enrollment in the study. The study with the use of patients' material was approved the Local Ethical Committee of the Privolzhsky Research Medical University (approval № 09 from 30.06.2023).
The study was conducted according to the guidelines of the Declaration of Helsinki and approved by the Local Ethical Committee of the Privolzhsky Research Medical University (approval № 09 from 30.06.2023). All surgery was performed under anesthesia, and every effort was made to minimize suffering.

Reviewer #1 (Public Review): https://doi.org/10.7554/eLife.94438.3.sa1
Reviewer #2 (Public Review): https://doi.org/10.7554/eLife.94438.3.sa2
Author response https://doi.org/10.7554/eLife.94438.3.sa3

## Additional files

### Supplementary files
• Supplementary file 1. Statistical significance of the differences of NAD(P)H a1-% between different cell lines and tumors (p-values).
• Supplementary file 2. Clinicopathological characteristics and NAD(P)H fluorescence decay parameters of patients' tumors.
• MDAR checklist

### Data availability
All data generated or analyzed during this study are included in the manuscript and supporting files. Source data files are provided for Figure 1, Figure 2, Figure 3. Raw FLIM data for cells and animal tumors are deposited in the BioStudies database (http://www.ebi.ac.uk/biostudies) under accession number S-BSST1536 (DOI:10.6019/S-BSST1536). The software is available at *Becker & Hickl, 2023*. Sharing restrictions are applied to the raw patients' data because original files may contain personal information. Processed versions of the dataset for the patients' files are provided as source data file for Figure 3 and Supplementary file 2.

The following dataset was generated:

| Author(s) | Year | Dataset title | Dataset URL | Database and Identifier |
|---|---|---|---|---|
| Komarova AD, Shirmanova MV | 2024 | Insights into metabolic heterogeneity of colorectal cancer gained from fluorescence lifetime imaging | https://doi.org/10.6019/S-BSST1536 | BioStudies, 10.6019/S-BSST1536 |

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
