## [Editor Report · eLife assessment]

This study presents a **valuable** finding on the heterogeneity of tumour metabolism using fluorescence lifetime imaging, measured across 4 cell lines, 4 tumour types of in vivo mouse models, and 29 patient samples. The indication is that the level of heterogeneity of cellular metabolism increases with model complexity and demonstrates high heterogeneity at a clinical level. The evidence supporting the claims of the authors is **solid**, and at the revision stage, the authors have included additional samples from 8 patients in the data pool, which is helpful for the conclusions that the authors are trying to draw. The work will be of interest to medical biologists developing methods for quantifying metabolic heterogeneity.

---

## [Referee Report · Reviewer #1 (Public Review)]

Summary:

In this study, Komarova et al. investigate the clinical prognostic ability of cell-level metabolic heterogeneity quantified via the fluorescence lifetime characteristics of NAD(P)H. Fluorescence lifetime imaging microscopy (FLIM) has been studied as a minimally invasive approach to measure cellular metabolism in live cell cultures, organoids, and animal models. Its clinical translation is spearheaded though macroscopic implementation approaches that are capable of large sampling areas and enable access to otherwise constrained spaces but lack cellular resolution for a one-to-one transition with traditional microscopy approaches, making the interpretation of the results a complicated task. The merit of this study primarily lies in its design by analyzing with the same instrumentation and approach colorectal samples in different research scenarios, namely in vitro cells, in vivo animal xenografts, and ex vivo tumor tissue from human patients. These conform to a valuable dataset to explore the translational interpretation hurdles with samples of increasing levels of complexity. For human samples, which exhibited the highest degree of heterogeneity from the experiments presented, the study specifically investigates the prediction ability of NAD(P)H fluorescence metrics for the binary classification of tumors of low and advanced stage, with and without metastasis, and low and high grade. They find that NAD(P)H fluorescence properties have a strong potential to distinguish between high- and low-grade tumors and a moderate ability to distinguish advanced stage tumors from low stage tumors. This study provides valuable results contributing to the deployment of minimally invasive optical imaging techniques to quantify tumor properties and potentially migrating into tools for human tumor characterization and clinical diagnosis.

Strengths:

The investigation of colorectal samples under multiple imaging scenarios with the same instrument and approach conforms to a valuable dataset that can facilitate interpretation of results across the spectrum of sample complexity.

The manuscript provides a strong discussion reviewing studies that investigated cellular metabolism with FLIM and the metabolic heterogeneity of colorectal cancer in general.

The authors do a thorough acknowledgement of the experimental limitations of investigating human samples ex vivo, and the analytical limitation of manual segmentation, for which they provide a path forward for higher throughput analysis.

Weaknesses:

NAD(P)H fluorescence provides a partial picture of the cell/tissue metabolic characteristics. Including fluorescence from flavins would comprise a more compelling dataset. These additional data should enable the quantification of redox metrics, which could positively contribute to the prognosis potential of metabolic heterogeneity. The authors did attempt to incorporate flavin fluorescence, unfortunately they could not find strong enough signal to proceed with the analysis.

---

## [Referee Report · Reviewer #2 (Public Review)]

Summary:

In the manuscript "Metabolic heterogeneity of colorectal cancer as a prognostic factor: insights gained from fluorescence lifetime imaging" by Komarova et al., the authors used fluorescence lifetime imaging and quantitative analysis to assess the metabolic heterogeneity of colorectal cancer. Generally, this work is logically well-designed, including in vitro and in vivo animal models and ex vivo patient samples. Although the key parameter (BI index) used in this study was already published by this group, it was shown that heterogeneity of patients' samples had associations with clinical characteristics of tumors. Additional samples from 8 patients were added to the data pool during the revision process, which is helpful and important for the conclusions that the authors are trying to draw. Overall, the revisions that the authors have made greatly strengthen this study.

Strengths:

(1) Solid experiments are performed and well-organized, including in vitro and in vivo animal models and ex vivo patient samples;

(2) Attempt and efforts to build the association between the metabolic heterogeneity and prognosis for colorectal cancer.

Weaknesses:

(1) Although additional data acquired from 8 patients were collected, maybe more patients should be involved in the future for reliable diagnosis and prognosis.

---

## [Author Response]

The following is the authors’ response to the original reviews.

**Public Reviews:**

**Reviewer #1 (Public Review):**
Summary:In this study, Komarova et al. investigate the clinical prognostic ability of cell-level metabolic heterogeneity quantified via the fluorescence lifetime characteristics of NAD(P)H. Fluorescence lifetime imaging microscopy (FLIM) has been studied as a minimally invasive approach to measure cellular metabolism in live cell cultures, organoids, and animal models. Its clinical translation is spearheaded through macroscopic implementation approaches that are capable of large sampling areas and enable access to otherwise constrained spaces but lack cellular resolution for a one-to-one transition with traditional microscopy approaches, making the interpretation of the results a complicated task. The merit of this study primarily lies in its design by analyzing with the same instrumentation and approach colorectal samples in different research scenarios, namely in vitro cells, in vivo animal xenografts, and tumor tissue from human patients. These conform to a valuable dataset to explore the translational interpretation hurdles with samples of increasing levels of complexity. For human samples, the study specifically investigates the prediction ability of NAD(P)H fluorescence metrics for the binary classification of tumors of low and advanced stage, with and without metastasis, and low and high grade. They find that NAD(P)H fluorescence properties have a strong potential to distinguish between high- and low-grade tumors and a moderate ability to distinguish advanced-stage tumors from low-stage tumors. This study provides valuable results contributing to the deployment of minimally invasive optical imaging techniques to quantify tumor properties and potentially migrate into tools for human tumor characterization and clinical diagnosis.Strengths:The investigation of colorectal samples under multiple imaging scenarios with the same instrument and approach conforms to a valuable dataset that can facilitate the interpretation of results across the spectrum of sample complexity.The manuscript provides a strong discussion reviewing studies that investigated cellular metabolism with FLIM and the metabolic heterogeneity of colorectal cancer in general.

The authors do a thorough acknowledgement of the experimental limitations of investigating human samples ex vivo, and the analytical limitation of manual segmentation, for which they provide a path forward for higher throughput analysis.

Weaknesses:To substantiate the changes in fluorescence properties at the examined wavelength range (associated with NAD(P)H fluorescence) in relationship to metabolism, the study would strongly benefit from additional quantification of metabolic-associated metrics using currently established standard methods. This is especially interesting when discussing heterogeneity, which is presumably high within and between patients with colorectal cancer, and could help explain the particularities of each sample leading to a more in-depth analysis of the acquired valuable dataset.

In order to address this issue, we have performed immunohistochemical staining of the available tumor samples for the two standard metabolic markers GLUT3 and LDHA.

The results are included in Supplementary (Fig.S4). Discussion has been extended.

Additionally, NAD(P)H fluorescence does not provide a complete picture of the cell/tissue metabolic characteristics. Including, or discussing the implications of including fluorescence from flavins would comprise a more compelling dataset. These additional data would also enable the quantification of redox metrics, as briefly mentioned, which could positively contribute to the prognosis potential of metabolic heterogeneity.

We agree with the Reviewer that fluorescence from flavins could be helpful to obtain more complete data on cellular metabolic states. However, we lack to detect sufficiently intensive emission from flavins in colorectal cancer cells and tissues. The paragraph about flavins was added in Discussion and representative images - in Supplementary Material (Figure S5).

In the current form of the manuscript, there is a diluted interpretation and discussion of the results obtained from the random forest and SHAP analysis regarding the ability of the FLIM parameters to predict clinicopathological outcomes. This is, not only the main point the authors are trying to convey given the title and the stated goals, but also a novel result given the scarce availability of these type of data, which could have a remarkable impact on colorectal cancer in situ diagnosis and therapy monitoring. These data merit a more in-depth analysis of the different factors involved. In this context, the authors should clarify how is the "trend of association" quantified (lines 194 and 199).

We thank the Reviewer for this suggestion. The section has been updated with SHAP analysis using different parameters (dispersion D of t2, a1, tm and bimodality index BI of t2, a1, tm). It is now more clear that D-a1 is more strongly associated with clinicopathological outcomes compared with other variables. We have also added some biological interpretation of these results in the Discussion.

**Reviewer #2 (Public Review):**
Summary:In the manuscript "Metabolic heterogeneity of colorectal cancer as a prognostic factor: insights gained from fluorescence lifetime imaging" by Komarova et al., the authors used fluorescence lifetime imaging and quantitative analysis to assess the metabolic heterogeneity of colorectal cancer. Generally, this work is logically well-designed, including in vitro and in vivo animal models and ex vivo patient samples. However, since the key parameter presented in this study, the BI index, is already published in a previous paper by this group (Shirshin et al., 2022), and the quantification method of metabolic heterogeneity has already been well (and even better) described in previous studies (such as the one by Heaster et al., 2019), the novelty of this study is doubted. Moreover, I am afraid that the way of data analysis and presentation in this study is not well done, which will be mentioned in detail in the following sections.Strengths:(1) Solid experiments are performed and well-organized, including in vitro and in vivo animal models and ex vivo patient samples.(2) Attempt and efforts to build the association between the metabolic heterogeneity and prognosis for colorectal cancer.Weaknesses:(1) The human sample number (from 21 patients) is very limited. I wonder how the limited patient number could lead to reliable diagnosis and prognosis;.

Additional 8 samples of patients’ tumors collected while the manuscript was under review were added to the present data. We agree that the number is still limited to conclude about the prognostic value of cell-level metabolic heterogeneity. But at this point we can expect that this parameter will become a metric for prognosis. We will continue this study to collect more samples of colorectal tumors and expand the approach to different cancer types.

(2) The BI index or similar optical metrics have been well established by this and other groups; therefore, the novelty of this study is doubted.

The purpose of this research was to quantify and compare the cellular metabolic heterogeneity across the systems of different complexity - commercial cell lines, tumor xenografts and patients’ tumors - using previously established FLIM-based metrics. For the first time, using FLIM, it was shown that heterogeneity of patients’ samples is much higher than of laboratory models and that it has associations with clinical characteristics of the tumors - the stage and the grade. In addition, this study provides evidence that bimodality (BI) in the distribution of metabolic features in the cell population is less important than the width of the spread (the dispersion value D).

Some corrections have been made in the text on this point.

**Recommendations for the authors:**

**Reviewer #1 (Recommendations For The Authors):**
The following comments should be addressed to strengthen the rigor and clarity of the manuscript.(1) The ethical committee that approved the human studies should also be mentioned in the methods section, as was done with the animal studies.

Information about the ethics committee has been added in the Manuscript.

The study with the use of patients’ material was approved by the ethics committee of the Privolzhsky Research Medical University (approval № 09 from 30.06.2023).

(2) The captions in Figures 2 and 3 must be revised. In Figure 2, it seems the last 2 sentences for the description of (C) do not belong there, and instead, the last sentence in the description of (D) may need to be included in (C) instead. Figure 3 is similar.

The captions were revised.

(3) From supplement Figure S2 it seems that EpCam and vimentin staining were only done in two of the mouse tumor types. No further mention is made in the results or methods section. Is there any reason this was not performed in the other tumor types? Were the histology and IHC protocols the same for the mouse and human tumors?

The data on other tumor types and patients’ tumors have been added in Figure S3. Discussion was extended with the following paragraph.

One of the possible reasons for metabolic heterogeneity could be the presence of stromal cells or diversity of epithelial and mesenchymal phenotypes of cancer cells within a tumor. Immunohistochemical staining of tumors for EpCam (epithelial marker) and vimentin (mesenchymal marker) showed that the fraction of epithelial, EpCam-positive, cells was more than 90% in tumor xenografts and on average 76±10 % in patients’ tumors (Figure S3). However, the ratio of EpCam- to vimentin-positive cells in patients’ samples neither correlated with D-a1 nor with BI-a1, which means that the presence of cells with mesenchymal phenotype did not contribute to metabolic heterogeneity of tumors identified by NAD(P)H FLIM.

(4) Clarify the design of the experiments: The results come from 50 - 200 cells in each sample (except 30 in the CaCo2 cell culture) that were counted from 5 - 10 images acquired from each sample. There were 21 independent human samples. How many independent samples were included in the cell culture experiments and the mouse tumor models? Why is there an order of magnitude fewer cells included in the CaCo2 group compared to the other groups (Figure 1)? From the image (Figure 1A - CaCo2), it seems to be a highly populated type of sample, yet only 30 cells were quantified. What prevents the inclusion of the same number of cells to be quantified in each group for a more systematic evaluation?

We thank the Reviewer for this comment.

Cell culture experiments included two independent replicates for each cell line, the data from which were then combined. In animal experiments measurements were made in three mice (numbered 1-3 in Figure 2C) for each tumor type. We have made calculations for additional >100 cells of CaCo2 cell line. In the revised version the number of Caco2 cells is 146.

The text of the Manuscript was revised accordingly.

(5) Regarding references: Some claims throughout the text would benefit from an additional reference. For example: line 70 "Metabolic heterogeneity [...] is believed to have prognostic value"; line 121 " [...] the uniformity of cell metabolism in a culture, which is consistent with the general view on standard cell lines [...]". The clinical translational aspect (i.e., paragraph in line 255) warrants the inclusion of the efforts already done with FLIM imaging in the clinical setting both in vivo and ex vivo with point-spectroscopy and macroscopy imaging (e.g., Jo Lab, Marcu Lab, French Lab, and earlier work by Mycek and Richards-Kortum in colorectal cancer to name a few).

Additional references were added.

**Reviewer #2 (Recommendations For The Authors):**
(1) In the Introduction, line 85, the authors mention that "Specifically, the unbound state of NAD(P)H has a short lifetime (~0.4 ns) and is associated with glycolysis, while the protein-bound state has a long lifetime (~1.7-3.0 ns) and is associated with OXPHOS". I do not think this claim is appropriate. One cannot simply say that the unbound state is associated with glycolysis, nor that the bound state is associated with OXPHOS; both unbound and bound state are associated with almost all the metabolic pathways. Instead, the expression of "glycolytic/ OXPHOS shift", as authors used in other sections of this manuscript, is a more appropriate one in this case.

The text of the Introduction was revised.

(2) What are the biological implications of the bimodality index (BI)? Please provide specific insights.

Bimodal distribution indicates there are two separate and independent peaks in the population data. In the metabolic FLIM data, this indicates that there are two sub-populations of cells with different metabolic phenotypes. Previously, we have observed bimodal distribution in the population of chemotherapy treated cancer cells, where one sub-population was responsive (shifted metabolism) and the second - non-responsive (unchanged metabolism) [Shirshin et al., PNAS, 2022]. In the naive tumor, a number of factors have an impact on cellular metabolism, including genetics features and microenvironment, so it is difficult to determine which ones resulted in bimodality. Our data on correlation of bimodality (BI) with clinical characteristics of the tumors show that there are no associations between them. What really matters is the width of the parameter spread in the population. The early-stage tumors (T1, T2) were metabolically more heterogeneous than the late-stage ones (T3, T4). A degree of heterogeneity was also associated with differentiation state, a stage-independent prognostic factor in colorectal cancer where the lower grade correlates with better the prognosis. The early-stage tumors (T1, T2) and high-grade (G3) tumors had significantly higher dispersion of NAD(P)H-a1, compared with the late-stage (T3, T4) and low-grade ones (G1, G2). From the point of view of biological significance of heterogeneity, this means that in stressful and unfavorable conditions, to which the tumor cells are exposed, the spread of the parameter distribution in the population rather than the presence of several distinct clusters (modes) matters for adaptation and survival. The high diversity of cellular metabolic phenotypes provided the survival advantage, and so was observed in more aggressive (undifferentiated or poorly differentiated) and the least advanced tumors.

The discussion has been expanded on this account.

(3) Have you run statistics in Figure 1B? If yes, do you find any significance? The same question also applies to Figures 2C and 3C.

We performed statistical analysis to compare different cell lines in in vitro and in vivo models, the results obtained are presented in Table S4.

(4) Line 119, why is the BI threshold set at 1.1?

When setting the BI threshold at 1.1, we relied on the work by Wang et al, Cancer Informatics, 2009. The authors recommended the 1.1 cutoff as more reliable to select bimodally expressed genes. Further, we validated this BI threshold to identify chemotherapy responsive and non-responsive sub-populations of cancer cells (Shirshin et al. PNAS, 2022)

(5) Line 123, what does the high BI of mean lifetime stand for? Please provide biological implications and insights.

The sentence was removed because inclusion of additional CaCo2 cells (n=146) for quantification NAD(P)H FLIM data showed no bimodality in this cell culture.

(6) In the legend for Figure 2C, the authors mention that "the bimodality index (BI-a1) is shown above each box"; however, I do not see such values. It is also true for Figure 3C.

The legends for Fig. 2 and 3 were corrected.

(7) In Figure 2, t1-t3 were not explained and mentioned in the main text. What do they mean? Do they mean different time points or different tumors?

t1-t3 means different tumors in a group. Changes have been made to the figure - individual tumors are indicated by numbers.

(8) In Figure 3, what do p13, p15 and p16 mean? It is not clearly explained. If they just represent patients numbered 13, 15, and 16, then why are these patients chosen as representatives? Do they represent different stages or are they just chosen randomly?

Figure 3 was revised. Representative images were changed and a short description for each representative sample was included. In the revised version, representatives have been selected to show different stages and grades.

(9) In Figure 3, instead of showing the results for each patient, I would suggest that authors show representative results from tumors at different stages; or, at least, clearly indicate the specific information for each patient. I do not think that providing the patient number only without any patient-specific information is helpful.

Figure 3 was revised.

(10) The sample number (21 patients) is very limited. I wonder how the limited patient number could lead to reliable diagnosis and prognosis.

Additional eight samples were added. The text, figures and tables were revised accordingly.

(11) In Discussion, it would be helpful to compare the BI index used in this study with the previously developed OMI-index (Line 275).

We believe that BI index and OMI index describe different things and, therefore, it is hard to compare them. While BI index is used to describe the degree of the metabolic heterogeneity, OMI index is an integral parameter that includes redox ratio, mean fluorescence lifetimes of NAD(P)H and FAD, and rather indicates the metabolic state of a cell. In this sense it is more relevant to compare it with conventional redox ratio or Fluorescence Lifetime Redox Ratio (FLIRR) (H. Wallrabe et al., Segmented cell analyses to measure redox states of autofluorescent NAD(P)H, FAD & Trp in cancer cells by FLIM, Sci. Rep. 2018; 8: 79). The assessment of the heterogeneity of the FLIM parameters has been previously reported using the weighted heterogeneity (wH) index Amy T. Shah et al, In Vivo Autofluorescence Imaging of Tumor Heterogeneity in Response to Treatment, Neoplasia 17, pp. 862–870 (2015). To the best of our knowledge, this is the only metric to quantify metabolic heterogeneity on the basis of FLIM data for today. A comparison of BI with the wH-index showed that the value of wH-index provides results similar to BI in the heterogeneity evaluation as demonstrated in our earlier paper E.A. Shirshin et al, Label-free sensing of cells with fluorescence lifetime imaging: The quest for metabolic heterogeneity, PNAS 119 (9) e2118241119 (2022). Yet, the BI provides dimensionless estimation on the inherent heterogeneity of a sample, and therefore it can be used to compare heterogeneity assessed by different decay parameters and FLIM data analysis methods. The limitation of using the OMI index for FLIM data analysis is the low intensity of the FAD signal, which was the case in our experiments.